# Validation of In Vitro Trained Transcriptomic Radiosensitivity Signatures in Clinical Cohorts

**DOI:** 10.3390/cancers15133504

**Published:** 2023-07-05

**Authors:** John D. O’Connor, Ian M. Overton, Stephen J. McMahon

**Affiliations:** Patrick G. Johnston Centre for Cancer Research, Queen’s University Belfast, Belfast BT9 7AE, UK; john.oconnor@qub.ac.uk

**Keywords:** radiosensitivity, transcriptomics, gene signatures

## Abstract

**Simple Summary:**

Radiation therapy (RT) is an important treatment for cancer. Advances in technology over the last 100 years have helped to deliver radiation to tumours (and avoid normal tissues) with increasing accuracy. Comparatively little progress has been made in adjusting radiation treatment based on genetic differences from one person to another. Techniques proposed for adjusting dose have been assessed in clinical datasets, but studies have not considered how technical aspects affect predictions or gene signature specificity to radiation. This work shows that preprocessing has a large influence on model predictions and demonstrates a lack of evidence for radiation specificity in existing gene expression-based models.

**Abstract:**

Transcriptomic personalisation of radiation therapy has gained considerable interest in recent years. However, independent model testing on in vitro data has shown poor performance. In this work, we assess the reproducibility in clinical applications of radiosensitivity signatures. Agreement between radiosensitivity predictions from published signatures using different microarray normalization methods was assessed. Control signatures developed from resampled in vitro data were benchmarked in clinical cohorts. Survival analysis was performed using each gene in the clinical transcriptomic data, and gene set enrichment analysis was used to determine pathways related to model performance in predicting survival and recurrence. The normalisation approach impacted calculated radiosensitivity index (RSI) values. Indeed, the limits of agreement exceeded 20% with different normalisation approaches. No published signature significantly improved on the resampled controls for prediction of clinical outcomes. Functional annotation of gene models suggested that many overlapping biological processes are associated with cancer outcomes in RT treated and non-RT treated patients, including proliferation and immune responses. In summary, different normalisation methods should not be used interchangeably. The utility of published signatures remains unclear given the large proportion of genes relating to cancer outcome. Biological processes influencing outcome overlapped for patients treated with or without radiation suggest that existing signatures may lack specificity.

## 1. Introduction

Radiation therapy (RT) is a key tool in modern cancer treatment [1,2]. Over the last century, physical refinements of radiation delivery have led to accurate targeting of tumours and improved sparing of normal tissues. However, biological refinement of RT has remained comparatively elusive. Mutations in single genes (e.g., ATM) which drastically alter cell sensitivity to radiation are rare and fail to explain the diversity in patient-to-patient response to ionising radiation [3]. Radiosensitivity prediction has instead focused on modelling genes that purportedly work in combination to determine in vitro or clinical responses to radiation (i.e., radiosensitivity signatures).

The development of pan-cancer radiosensitivity signatures has utilized microarray transcriptomics in cancer cell lines alongside radiosensitivity measurements from in vitro assays (e.g., clonogenic survival) to identify genes that correlate with cell response to radiation treatment. Clinical data has also been used directly to build radiosensitivity signatures for breast and prostate cancer [4,5,6]. Although pan-cancer models trained on in vitro radiation-specific assays may provide superior generalizability and specificity than clinical models, robust analysis of the accuracy of such preclinical models has not yet been performed.

The radiosensitivity index (RSI) is a 10-gene signature that was developed using a linear model of ProbeSets from the Affymetrix HU6800 microarray to predict SF2 with the covariates of tissue-of-origin, ras status and p53 status [7]. The 500 top-ranked genes were taken to build a gene expression network from which hubs were identified. The expression ranking for the hubs was input for training a linear regression model. Despite poor in vitro validation performance in the initial publication of the RSI and in subsequent independent studies [7,8,9], clinical studies have correlated cancer outcome and RSI [10,11,12,13,14,15,16]. The Kim signature [17] is made up of 31 genes, which were identified by significance analysis of microarrays on four different platforms. Although a principal components analysis appeared to show separation between radioresistant, intermediate, and radiosensitive groups, no predictive model has been published, and the Kim signature has not been tested in clinical data. The RadSigBench framework found that both the RSI and Kim signatures were unable to predict in vitro radiosensitivity with greater accuracy than randomly resampled control genes [9].

Several factors may contribute to differences between in vitro and clinical results. An important, often overlooked, element is the confounding of the relationship between genomics and clinical outcome by general biological processes (e.g., proliferation) as opposed to those of specific interest (e.g., intrinsic radiosensitivity). Previous work has shown that associations between gene expression signatures and outcome in breast cancer are largely abrogated after adjusting for proliferation [18]. The same study showed that 90% of signatures made up of >100 randomly sampled genes were significantly correlated with breast cancer outcome [18]. The application of microarray trained signatures in clinical data may also be unreliable due to technical issues, such as the type of normalisation used and the method for summarising ProbeSets to gene-level data. For example, the RSI model was trained on MAS5 normalised data but has been implemented using other normalisation algorithms, such as RMA [19].

We investigate the application of in vitro radiosensitivity signatures to publicly available clinical datasets. Signatures were tested for association with time to first recurrence and overall survival, comparing results with resampled control signatures. Single gene models were examined in order to assess the correlation between individual gene expression and outcome as well as investigating the biological processes related to RT. The effects of different normalisation methods upon radiosensitivity estimation were also evaluated.

## 2. Materials and Methods

Clinical and microarray data were obtained from four cohorts: the glioblastoma (GBM) dataset from The Cancer Genome Atlas (TCGA) [20], lung adenocarcinoma (LUAD) data from GSE68465 [21], and the Erasmus & Karolinska breast cancer cohorts [22,23] (Table 1). Datasets were included if they had expression from a microarray with the RSI ProbeSets [7] and Kim signature genes [17], as well as an indicator of whether radiation treatment was delivered and survival/recurrence information. Molecular data for the GBM dataset was downloaded from the TCGA and matched with clinical data from [16]. Microarray data for the Karolinksa database was downloaded from the Gene Expression Omnibus (GEO) database (GSE1456) along with the Erasmus dataset (GSE2034 & GSE5327) and the LUAD cohort (GSE68465). Data for the NCI60 was also obtained from the GEO database (GSE32474). The TCGAbiolinks (v2.24.3) and GEOquery (v2.64.2) packages in R 4.2.1 [24] were used to access these datasets [25,26].

Where microarray data is used, there is a need to remove non-biological sources of noise from the data. This is often achieved using an algorithm for identifying and eliminating biases. Three algorithms that have been used for this purpose are: RMA (Robust multiarray average), MAS5 (Microarray analysis suite 5.0) and IRON (Iterative Rank Order Normalisation). The use of rankings of genes after normalisation (as opposed to raw expression values) in the RSI model may help to negate the differences arising from the application of different normalisation approaches, but this assumption has not been tested despite all three algorithms being used for training and application in publications [15]. The RSI model was trained on MAS5 normalised NCI60 data [7]. Here, we calculated the RSI score value in clinical data normalised using MAS5, RMA, and IRON. The limits of agreement (the average difference ±1.96 SD of the differences) between the RSI values for each of the outcomes was assessed [27]. Normalisation was performed using the affy (v1.74.0) package from Bioconductor [28] in R 4.2.1 [24] for RMA/MAS5 and libaffy (v2.2.0) (http://gene.moffitt.org/libaffy/ (accessed on 21 July 2022)) for IRON [29].

Previous clinical studies have shown that a large portion of the transcriptome is related to cancer outcomes, leading to a high probability that a randomly sampled signature will correlate with outcome [18]. This baseline clinical correlation, coupled with the poor predictive capacity of radiosensitivity signatures in vitro [9], provides impetus for benchmarking signatures of radiosensitivity in clinical data against resampled gene sets with no biological rationale to test for radiation-specific predictive ability. Resampled signatures were constructed to match the modelling process of the published RSI and Kim signatures. The RSI model is a linear regression on the rankings of 10 genes. We generated 500 SigResamp10 signatures by performing the same linear regression on resampled sets of 10 genes (Appendix A). The Kim study reported 31 genes and performed a principal components analysis but did not report a predictive model. Principal components regression was therefore fitted for both the Kim model and the 500 resampled equivalent signatures (SigResamp31) (Appendix A). The expression of MGMT (ProbeSet ID: 204880_at) was added as a covariate in overall survival analyses for GBM to match previous clinical validation of RSI [16]. All models were trained using NCI60 data (GSE32474) and previously published SF2 values [7]. Radiosensitivity values were standardised to z-scores for survival analysis by subtracting the mean and dividing by the standard deviation. Cox regression models were fitted using the ‘cph’ function in the rms (v6.3-0) package in R [30].

Along with radiosensitivity predictions from combinations of genes in signatures, univariate Cox regression models were investigated for all genes on the clinical microarrays. Cell processes, such as proliferation, involve changes in a substantial proportion of the transcriptome and can influence the progression and effectiveness of treatment in cancer. Univariate gene models allow calculation of the hazard ratio (i.e., recurrence or survival) associated with increased expression of a given gene for different treatments (i.e., RT or no RT). The distribution of *p*-values from univariate Cox regression analysis was tested against a uniform distribution (which is expected under the null hypothesis [31]) using the Kolmogorov–Smirnov test in the dgof (v1.4) R package [32] in order to identify where the *p*-value distribution deviates from the expectation, particularly where there are more significant genes (<0.05) than expected by chance. The single gene model hazard ratios were also used to assess the biological process involved in outcomes using functional annotation.

Overrepresentation analysis (Appendix A) was used to test if genes in published signatures were indicative of biological processes defined by the Gene Ontology. This analysis was performed using the ‘enrichGO’ function from the clusterProfiler (v4.4.4) Bioconductor package in R [33]. *p*-values from enrichment tests were adjusted by the Benjamini–Hochberg method [34]. We also required at least two genes in common between the signature and identified pathway to reduce false positives arising from small signatures (i.e., only 1 out of 10 genes being present in the pathway).

Gene set enrichment analysis (GSEA) is often used to analyse ranked lists of fold changes for genes between two phenotypes; GSEA allows for the identification of biological processes that are differently expressed between the conditions studied. Instead of examining relative expression between groups, here we use the ranked list of hazard ratios (from univariate survival analysis) for all genes on the microarrays (Appendix A). This approach enables identification of biological processes that are overrepresented at the top/bottom of the ranked hazard ratio list and are thus correlated with the clinical endpoint studied. Datasets were pooled by outcome (i.e., first recurrence or overall survival) and stratified by cohort in cox regression. Patients were also separated by treatment (i.e., whether they received radiation or not). This resulted in 4 groups: FR_RT (patients with first recurrence as the outcome who received RT), FR_noRT, OS_RT and OS_noRT. This analysis was performed using the ‘gseGO’ function from clusterProfiler (v4.4.4) [33]. Reduction of gene ontology terms and visualisation was performed using the *rrvgo* (v1.8.0) with a similarity threshold of 0.7 for both overrepresentation and GSEA analysis [35]. The semantic similarity was calculated within the rrvgo R package using the function ‘GOSemSim’ [36]. GSEA analysis (Appendix A) was also performed on gene-by-gene differences in hazard ratio between RT and noRT groups for both overall survival and first recurrence (Appendix A).

## 3. Results

RSI values were calculated using three different normalisation algorithms on identical clinical data. Although mean differences between the methods are small (range: 0.02 to 0.06), the 95% limits of agreement were −0.22–0.16 for MAS5/RMA, −0.26–0.14 for MAS5/IRON and −0.16–0.21 for IRON/RMA (Figure 1). These results suggest that discrepancies of >20% should be expected when using one normalisation method for training a model and another in implementation, as is often the case in the literature [15]. Estimations of model accuracy from validation studies may not hold if different normalisation algorithms are used in practice.

The Kim et al. signature had a hazard ratio significantly different from 1 in the FR_noRT group with a 53% (95% CI: 6–120%) increase per SD and in the OS_RT group with a 24% (95% CI: 7–44%) increase per SD (Figure 2). The RSI signature’s relative hazard ratio was not significantly different from 1 for any of the groups examined. The mean hazard ratio for resampled 10-gene and 31-gene signatures was within the 95% CI of their published counterparts for all outcome and treatment combinations (Figure 2). Interestingly, all resampled signatures except SigResamp10 in the OS_noRT group) had hazard ratios significantly greater than 1 (Figure 2). The effect sizes were small (between 2% and 7%) in most cases except in the FR_noRT group where a 1 SD increase in SigResamp31 (mean accuracy) signature value was associated with a 17% (95% CI: 15–19%) increase in relative hazard (Figure 2). The proportion of models with a *p*-value of less than 0.05 increased with signature size; 15.0% of the 500 resampled 10-gene signatures were significant compared to 19.5% for the 100-gene signatures (Appendix A).

Overrepresentation analysis identified actin filament organization, cell substrate adhesion, and membrane raft organisation among 20 clusters from 74 biological processes overrepresented in the Kim signature (Figure 3). A total of 259 biological processes were overrepresented in the RSI signature (Figure 3). These were reduced to 39 clusters, including the response to reactive oxygen species, regulation of NF-κB signalling, response to mechanical stimulus, response to cadmium ion, and muscle cell proliferation (Figure 3).

The distribution of *p*-values from fitting Cox regression models to every gene from the microarray data were tested for differences from the null (i.e., uniform) distribution. Distributions for all groups were significantly (*p* < 0.001) different from uniform, with the peak occurring in the <0.05 range, which suggests that many more genes were related to cancer outcome than expected by chance (Figure 4, Figure 5, Figure 6 and Figure 7). The proportion of genes with a *p*-value of less than 0.05 was 18%, 17%, 13%, and 12% for the FR_noRT, FR_RT, OS_noRT, and OS_RT groups respectively.

Gene set enrichment analysis was performed on four lists of genes ranked by univariate hazard ratio; the groups were pooled by outcome (i.e., overall survival or recurrence) and divided based on treatment (i.e., radiation treated or not). Significant pathways were clustered by semantic similarity and the top 10 clusters were visualised. Genes that were overrepresented at the top (or bottom) of the ranked list of univariate hazard ratios for FR_noRT were involved in nuclear chromosome segregation, cell cycle DNA replication, chromosome organization, and cell division, which suggests that genes involved in the proliferation processes were indicative of first recurrence risk (Figure 4). Several proliferation-related biological processes were also present in the FR_RT GSEA results, with chromosome organization, mitotic sister chromatid segregation, DNA replication, and cell division all in the top 10 clusters (Figure 5). FR_RT also contained multiple pathway clusters involved in the immune response, including the leukocyte proliferation and positive regulation of cell activation (Figure 5).

Genes involved in proliferation and immune response were also enriched in the OS_noRT group; the top clusters included chromosome organization, regulation of DNA templated DNA replication, mitotic cell cycle checkpoint signalling, and adaptive immune repsonse (Figure 6). Analysis of OS_RT found processes responsible for migration and locomotion (Figure 7), which could impact overall survival due to difficulty in targeting cancer cells, since radiation is targeted at the primary tumour and these processes are important for metastasis.

## 4. Discussion

Radiation has been a mainstay in cancer treatment for over a century but still lacks biological personalisation. Attempts have been made to predict an individual’s intrinsic sensitivity to radiation based on gene expression data and laboratory tests of radiosensitivity. These models have proven inaccurate when independently tested on held out in vitro data, but clinical studies have shown association with outcome [9]. Here, we assessed the influence of input data normalisation on predictions of radiosensitivity. Since many biological processes affect both gene expression and outcomes following cancer treatment, we also sought to investigate the predictive capacity of published radiosensitivity signatures compared to random sets of genes. Functional annotation was utilised to identify biological processes associated with the random (resampled) gene signatures and individual genes that performed well in predicting overall survival and recurrence.

The normalisation algorithm choice influenced radiosensitivity predictions. The limits of agreement for different normalisation algorithms were wide for the RSI signature, suggesting that calculations using different methods are not equivalent and should not be used interchangeably. The RSI estimates survival fraction, which theoretically ranges from 0 to 1, and the limits of agreement exceeded 0.2 in all comparisons, indicating that a difference of >20% of the range can result from data normalisation choice alone. When applied in clinical data, this translated into a difference in the estimated hazard ratio of up to 0.25, significantly impacting on the apparent performance of signatures (Appendix A). Of course, differing normalisation methods are expected to produce different output values, but our work exposes the flawed assumption that the ranking of the genes (as used in the RSI model) would be unaffected. Given the potentially large difference in the estimated radiosensitivity and the relative technical ease of switching normalisation algorithms, we show that the same normalisation method should be used in training and implementation of radiosensitivity signatures. This is important as many published works have used different normalisation algorithms interchangeably [7,15,16].

The published models were tested using a subset of publicly available data from the latest validation of the RSI model (in GARD form) [16] and an additional lung cancer cohort [21]. The results of testing against resampled signatures are in line with previously published in vitro analyses and suggest that the confidence interval for the performance of both RSI and Kim signatures included the mean accuracy resampled signature [8,9]. While further optimisation of models may be possible, our work sought to use model structures from published signatures to address non-specific correlations across the transcriptome and with cancer outcomes. The framework described here can also be used to test other models and provide evidence for (or against) direct involvement in the molecular mechanisms controlling radiation response.

Analysis of single genes’ correlation with cancer outcome showed around 18% of all genes have a *p*-value of less than 0.05 in a Cox regression for recurrence, while around 12% are significant in overall survival models. Given that multiple genes are typically combined in radiosensitivity signatures, there is a tangible risk of developing a signature that significantly correlates with outcome without necessarily being specific to the process of interest, such as radiation response. This is evidenced by 15% of resampled 10-gene signatures being significant, with the proportion increasing to 19.5% for 100-gene signatures (Appendix A).

Although the RSI model did not show an improvement over resampled signatures, overrepresentation analysis showed that, for the 10 genes in the model, biological processes previously associated with radiosensitivity were identified, including response to reactive oxygen species (which are created by ionising radiation) and immune regulation. A recent study has shown that the RSI is associated with immune response in a large sample of tumours, which is not unexpected given the inclusion of STAT1 and IRF1 [37]. However, other processes that do not obviously align with radiation response were identified, such as response to cadmium ion, response to mechanical stress, and gland development. RSI was trained by selecting highly connected hubs of a large network and does not accurately predict SF2 in vitro; in light of this and the data presented in our work, we would suggest that the signature may not necessarily be specific to radiation and is indicative of multiple processes that may influence cancer outcomes, such as proliferation, response to other treatments, or cancer progression.

Enrichment analysis with hazard ratios from single gene models suggested that immune response is important across multiple groups and in cohorts that did not receive radiation. Immune processes are increasingly being recognised as crucial to cancer progression and treatment response [38]. Proliferation was also an important factor in single genes’ relationship to both recurrence and overall survival. Migration and locomotion processes were additionally identified in OS_RT, which may be related to radiation response through lack of proliferation in migrating cells and consequent lower proportion of cells in S-phase [39]. Overall, many biological processes were identified as being associated to cancer outcome in these datasets and further work is required to ensure specificity to radiation. This emphasises the need for more stringent false positive control and external validation of signatures.

Future development of radiation response signatures may include multiple genomic data types that are already publicly available, such as, for example, NCI60 [40] and CCLE [41]. There may be benefits in using RNAseq instead of microarray expression (e.g., for better identification of isoforms), and incorporation of mutational, proteomic and post-transcriptional modification data. These genomic data types taken together may provide a more reliable indication of gene functional status given that mutation and expression information may not propagate to the protein level [42]. Perturbation data may also be important in assessing the genomic response to radiation and improving predictions [43]. The relationship between genetic factors and radiosensitivity may be dependent on clinical (e.g., sex and age) and microenvironmental (e.g., hypoxia) factors, which can be incorporated into models [44,45]. The SF2 value from the clonogenic assay has been the foundation of many in vitro trained models and has been used as a gold standard in the development of new radiosensitivity assays [46], despite showing mixed results in association with clinical outcome [47,48] and a relatively poor reproducibility [49].

## 5. Conclusions

Our work has shown that future studies of radiosensitivity prediction should implement more intensive validation in the model development stage to help ensure wide applicability and progression to clinical use. In particular, benchmarking of signatures against collections of resampled genes allows for contextualisation of prediction errors. Other sources of variation between training and application of models, such as normalisation, should also be quantified in the model development phase. Robust error estimation, as we outline here, may be incorporated into radiosensitivity signature training, improving specificity, and helping to drive advances in biological personalisation of radiation therapy.

## Figures and Tables

**Figure 1 cancers-15-03504-f001:**
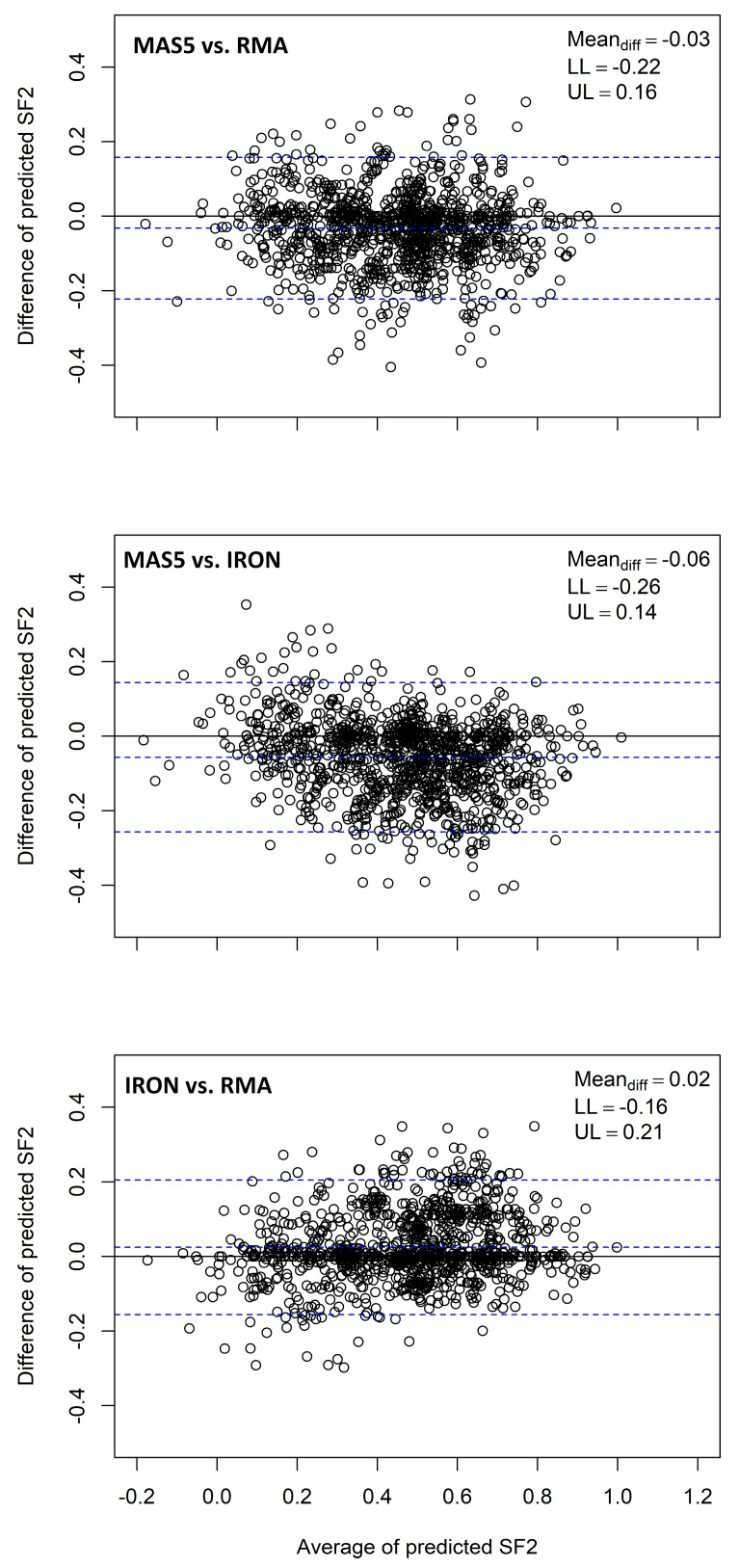
Bland–Altman plots for calculation of RSI (unstandardised values) from clinical data (n = 1173, the total number of patients with RSI calculated) using three normalisation algorithms (MAS5, RMA, and IRON). The limits of agreement are around ±20% of the range for the RSI (i.e., 0 to 1), suggesting the normalisation methods should not be used interchangeably.

**Figure 2 cancers-15-03504-f002:**
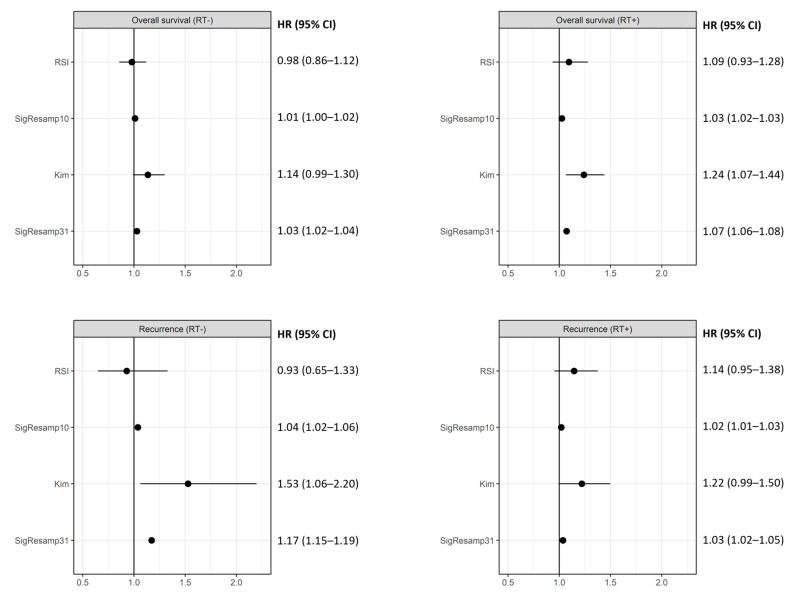
Results from pooled Cox regression for RSI and Kim models as well as mean accuracy of randomly sampled signatures with 10 and 31 genes (SigResamp10 and SigResamp31). The RSI was not significant with any outcome or treatment whereas the Kim signature was related to recurrence without radiation and overall survival with radiation. The mean for randomly sampled signatures was within the 95% CI for both the RSI and Kim signatures.

**Figure 3 cancers-15-03504-f003:**
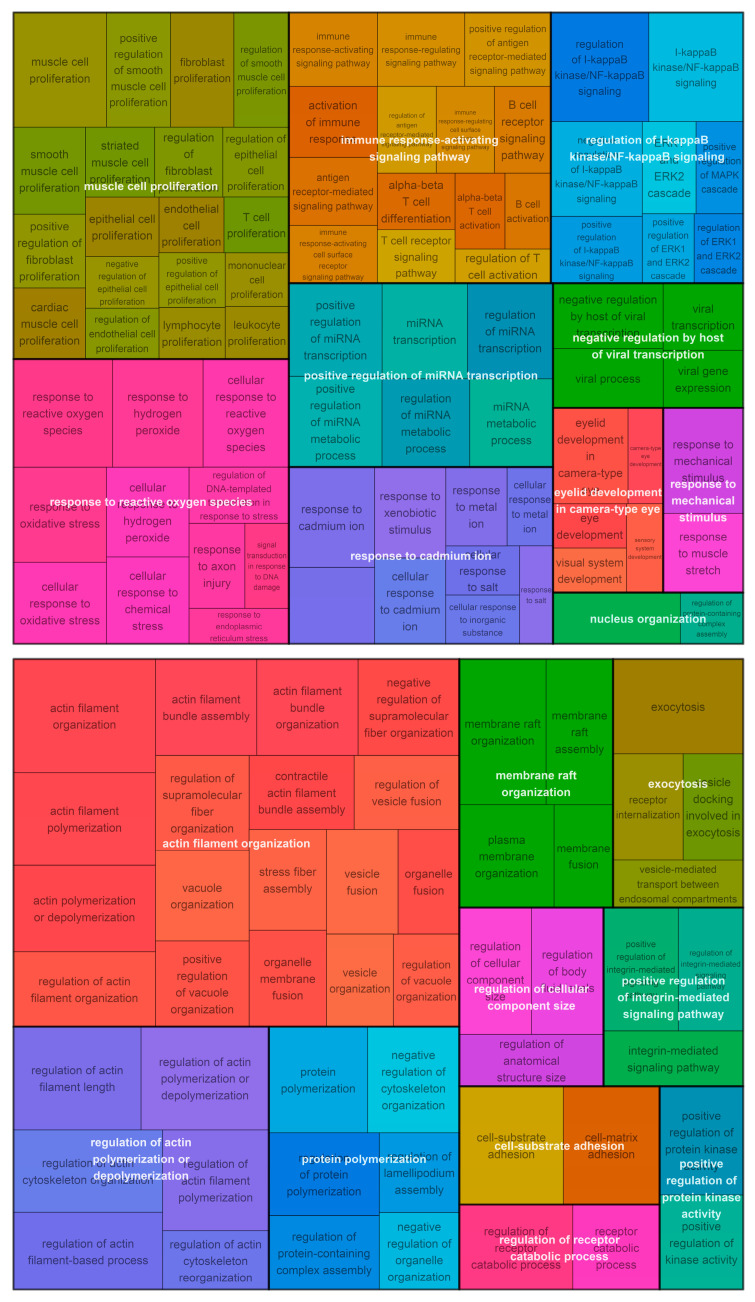
Biological pathways overrepresented in the RSI (**top**) and Kim (**bottom**) signature genes. Biological processes traditionally associated with response to radiation (e.g., response to reactive oxygen species) were identified in the RSI signature along with immune response. Processes related to proliferation were enriched for both signatures.

**Figure 4 cancers-15-03504-f004:**
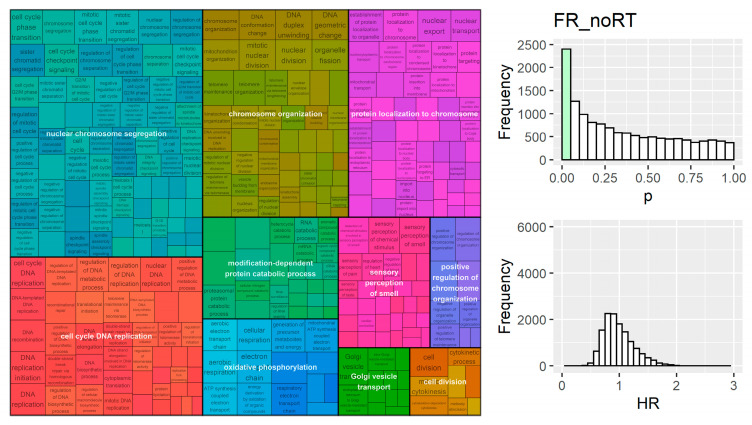
Results from GSEA on univariate gene models (**left**) ranked by their hazard ratio in survival analysis for the FR_noRT group. Proliferation related processes were evident. The distribution of univariate *p*-values for all genes tested (**right-top**) was biased towards lower values (*p* < 0.001 for deviation from uniform distribution).

**Figure 5 cancers-15-03504-f005:**
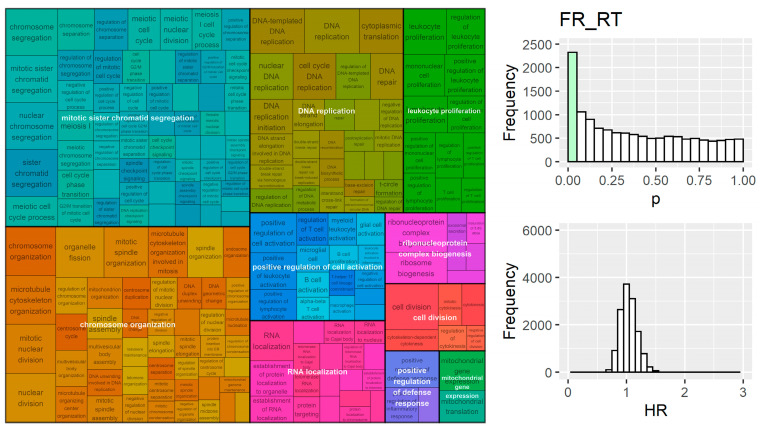
Results from GSEA on univariate gene models ranked by their hazard ratio in survival analysis for the FR_RT group. Proliferation and immune processes were overrepresented at the top/bottom of the univariate hazard ratio list in this radiation treated group. The distribution of univariate *p*-values for all genes tested (**right-top**) was biased towards lower values (*p* < 0.001 for deviation from uniform distribution).

**Figure 6 cancers-15-03504-f006:**
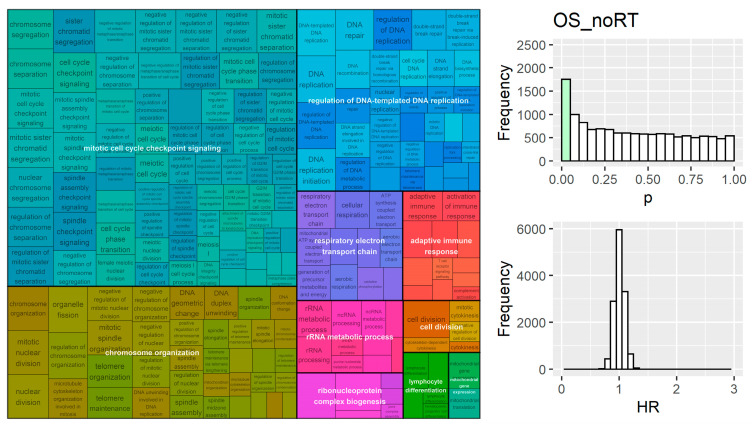
Results from GSEA on univariate gene models ranked by their hazard ratio in survival analysis for the OS_noRT group. Immune and proliferation processes were significant. *p*-values for genes were biased towards lower values (*p* < 0.001 for deviation from uniform distribution).

**Figure 7 cancers-15-03504-f007:**
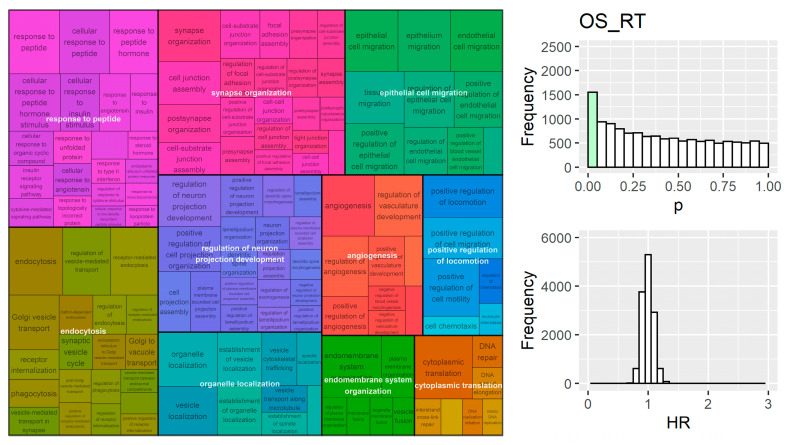
Results from GSEA on univariate gene models ranked by their hazard ratio in survival analysis for the OS_RT. Migration and metastasis related processes were significant. The distribution of univariate *p*-values for all genes tested (**right-top**) was biased towards lower value (*p* < 0.001 for deviation from uniform distribution).

**Table 1 cancers-15-03504-t001:** Cohorts included in groups along with the sample size (*n*) and number of events.

Group	Datasets	*n*	Events
Overall Survival (RT+) [OS_RT]	TCGA glioblastoma	186	133
	Karolinska breast cancer cohort	77	17
	LUAD cohort	65	51
Overall Survival (RT-) [OS_noRT]	TCGA glioblastoma	55	55
	Karolinska breast cancer cohort	82	23
	LUAD cohort	364	174
Recurrence (RT+) [FR_RT]	Erasmus breast cancer cohort	282	91
	Karolinska breast cancer cohort	77	19
Recurrence (RT-) [FR_noRT]	Erasmus breast cancer cohort	62	12
	Karolinska breast cancer cohort	82	21

## Data Availability

All data is available from public datasets identified in the main text.

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
