# Peer review of "Validation of In Vitro Trained Transcriptomic Radiosensitivity Signatures in Clinical Cohorts"

_cancers, 2023, doi:10.3390/cancers15133504_

Round 1

Reviewer 1 Report

Thank your for the opportunity to review the manuscript of O'Connor et al. Transcriptomic individualisation of radiation oncology is an important topic where more progress needs to be made to translate in the clinic.

The introduction to the article focusses on the radiosensitivity index and the Kim signature. However there are other signatures of radiosensitivity which I think should be mentioned  such as the  mRNA signature of breast cancer radio sensitivity described by Speers et al Clin Cancer Res 2015;21:3667-77 and ARTIC  Sjostrom M et al J Clin Oncology 2019;37:3340-2019.

Author Response

Thank you to the reviewer for the assessment of our work.

We have focused on in vitro trained signatures in our work, but the reviewer is correct that many tissue-specific clinical models exist and have shown promising results. We have now signposted this in the introduction and put forward our reasons for analysing pan-cancer, in vitro trained models.

“Clinical data has also been used directly, to build radiosensitivity signatures for breast and prostate cancer [4-6]. Although pan-cancer models trained on in vitro radiation-specific assays may provide superior generalizability and specificity than clinical models, robust analysis of the accuracy of such preclinical models has not yet been performed.” 

Reviewer 2 Report

Great work with a significant contribution to the field of personalized medicine. The data analysis is thorough and well-presented, allowing for clear interpretation of the findings. Kudos to the authors. 

Some minor editing of the English language is needed.

Author Response

Thank you to the reviewer for the positive assessment of our work. 

Reviewer 3 Report

Personalized therapeutic treatment is the future strategy against cancer. It is an important and interesting topic. However, I don’t think the authors present the data clearly. The followings are some concerns and comments have been pointed out that the authors may want to consider.

1.      Line 29: Please define the abbreviation before using it. For example, “RT (radiation therapy)”.

2.      Lines 39-51: Please include the necessary references.

3.      Line 177 Figure 1 related: Please a) provide high-resolution images; b) provide clear explanations. For example, where did n=1173 come from, and so on?

4.      Line 210: Please use italic p as it refers to a p-value throughout the manuscript.

5.      Where are the descriptions of the results for Figure 1 to Figure 7?

6.      Please describe your results clearly, from Figure 1 to Figure 7, instead of general summarization.

7.      Please check references seriously throughout the manuscript. There are lots of “Error! Reference source not found.”.

Author Response

Thank you to the reviewer for the assessment of our manuscript. Please find below changes and explanations.

  1. We have now defined RT in the first line of the Simple Summary. A number of other abbreviations are also now defined at first usage.

  1. Citations have been added to the first paragraph of the introduction as requested

  1. A) All figures have been replaced with higher resolution images B) we have clarified in the caption where the sample size comes from for the RSI repeatability, figure 1, comes from

“Figure 1. Bland-Altman plots for calculation of RSI (unstandardised values) from clinical data (n=1173, the total number of patients with RSI calculated) using three normalisation algorithms (MAS5, RMA and IRON). The limits of agreement are around ±20% of the range for the RSI (i.e., 0 to 1) suggesting the normalisation methods should not be used interchangeably. “

  1. All instances of “p-value” have been replaced with “p-value” and p with p

5/6/7. Unfortunately, in the previous submission the figure references were corrupted during uploading of the manuscript and replaced with “Error! Reference source not found.”. We apologise for this and have amended the results section to fix it. The descriptions in the results section are now correctly linked to the relevant figures and discusses them in sequence as recommended.